# Loss of cortical control over the descending pain modulatory system determines the development of the neuropathic pain state in rats

Robert AR Drake[1]*, Kenneth A Steel[2], Richard Apps[1], Bridget M Lumb[1], Anthony E Pickering[1,3]

[1]School of Physiology, Pharmacology & Neuroscience, University of Bristol, Bristol, United Kingdom; [2]School of Biosciences, University of Cardiff, Cardiff, United States; [3]Bristol Anaesthesia, Pain & Critical Care Sciences, Bristol Medical School, Bristol Royal Infirmary, Bristol, United Kingdom

**Abstract** The loss of descending inhibitory control is thought critical to the development of chronic pain but what causes this loss in function is not well understood. We have investigated the dynamic contribution of prelimbic cortical neuronal projections to the periaqueductal grey (PrL-P) to the development of neuropathic pain in rats using combined opto- and chemogenetic approaches. We found PrL-P neurons to exert a tonic inhibitory control on thermal withdrawal thresholds in uninjured animals. Following nerve injury, ongoing activity in PrL-P neurons masked latent hypersensitivity and improved affective state. However, this function is lost as the development of sensory hypersensitivity emerges. Despite this loss of tonic control, opto-activation of PrL-P neurons at late post-injury timepoints could restore the anti-allodynic effects by inhibition of spinal nociceptive processing. We suggest that the loss of cortical drive to the descending pain modulatory system underpins the expression of neuropathic sensitisation after nerve injury.

**\*For correspondence:**
robert.drake@bristol.ac.uk

**Competing interests:** The authors declare that no competing interests exist.

## Introduction

There is a pressing need to better understand the causal mechanisms of chronic pain and develop effective therapeutic strategies that will alleviate its societal burden (*Breivik et al., 2006*). The brain, as opposed to the periphery, has received increasing focus as a critical contributor to chronic pain development (*Ossipov et al., 2010*; *Denk et al., 2014*; *Baliki and Apkarian, 2015*). The descending pain modulatory system (DPMS) links brain and spinal cord to provide potent and targeted regulation of nociceptive processing at multiple levels of the neuroaxis, including the spinal dorsal horn (*Millan, 2002*; *Tracey and Mantyh, 2007*). Importantly, the DPMS can affect the perception of pain and is a critical regulator of the development of the pain state following injury (*Eippert et al., 2009*; *Hughes et al., 2013*; *Drake et al., 2014*; *Hirschberg et al., 2017*).

Typically, following acute injury, this descending regulation functions to inhibit spinal dorsal horn circuits that subserve damaged tissue and, in doing so, moderate central sensitisation (*Vanegas and Schaible, 2004*; *Drake et al., 2014*). However, net loss of inhibitory control has been noted in a wide variety of human chronic pain disorders, and descending inhibitory systems are depleted and non-functional in animal models of persistent pain (*Yarnitsky, 2010*; *Hughes et al., 2013*; *Hughes et al., 2015*; *Staud, 2012*; *Bannister et al., 2015*). Similarly, trait deficiencies in endogenous inhibitory control and/or its engagement by peripheral injury are thought to impart individual vulnerability to chronic pain (*Edwards, 2005*; *Yarnitsky, 2010*; *Granovsky, 2013*; *Denk et al., 2014*; *González-Roldán et al., 2020*). What causes this deficit/loss in the function of the DPMS is

not well understood but could help identify critical and generalisable mechanisms of chronic pain development that lay the foundation for the development of more effective therapeutic strategies.

In humans, the medial prefrontal cortex (mPFC) displays specific activity related to acute pain processing, pain expectation, and endogenous pain modulation (*Lorenz et al., 2002*; *Wager et al., 2004*; *Wiech and Tracey, 2009*; *Legrain et al., 2011*; *Brooks et al., 2017*). Importantly, the mPFC shows alterations in structure and function that are related to and, sometimes, predictive of the transition to chronic pain (*Apkarian et al., 2004*; *Baliki et al., 2006*; *Baliki et al., 2012*). Direct corticofugal projections from the mPFC to the midbrain link it to the DPMS to provide a route to pain state regulation (*An et al., 1998*; *Huang et al., 2019*). The midbrain periaqueductal grey (PAG) is a core component of the DPMS able to facilitate and/or inhibit spinal nociceptive processing via pain modulatory brainstem nuclei including the rostral ventromedial medulla and locus coeruleus (*Mantyh, 1983*; *Waters and Lumb, 2008*; *Ossipov et al., 2010*; *Drake et al., 2016*). Notably, altered functional connectivity between the mPFC and PAG is observed in human patients with musculoskeletal, neuropathic, and inflammatory chronic pain, suggesting that altered cortical control may contribute to maladaptation of the DPMS and that this mechanism may be relevant to chronic pain in general (*Cifre et al., 2012*; *Yu et al., 2014*; *Chen et al., 2017*; *Mills et al., 2018*; *Segerdahl et al., 2018*).

Recently, preclinical investigation has demonstrated the prelimbic (PrL) cortex, a division of the rodent mPFC, is able to affect noxious thresholds in neuropathic rats (*Huang et al., 2019*). However, whether the contributions of PrL neurons that target the PAG (PrL-P) in sensory and/or affective aspects of the pain state are dynamically altered during the development of neuropathic pain is not known. To assess this question, we transfected PrL-P neurons with excitatory optogenetic and inhibitory chemogenetic actuator proteins to allow selective manipulation of their activity (*Zhang et al., 2010*; *Sternson and Roth, 2014*). This enabled their contribution to sensory and affective aspects of pain-like behaviour to be charted before and, at regular intervals, following peripheral nerve injury in rats. We also used electrophysiological methods to investigate whether PrL-P neurons exert effects on spinal dorsal horn nociceptive circuit activity to establish whether these effects are mediated by descending control.

## Results

### Targeting mPFC → PAG neurons in the PrL cortex

To make selective manipulations of mPFC neurons that project to the PAG, we expressed the excitatory light-activated ion channel, channelrhodopsin-2 (ChR2), and the inhibitory ligand gated G protein–coupled receptor, hM4Di, in mPFC pyramidal neurons using an intersectional and Cre-dependent approach (*Figure 1A*). This approach led to the expression of hM4Di-mCherry and/or Chr2-YFP on average in 248±71 mPFC pyramidal neurons (n=3 rats) that were located in layer 5/6 (*Figure 1B and C*). Colocalisation of hM4Di-mCherry and Chr2-EYFP was found in 76.1±3.3% of labelled neurons with 23.9±3.3% expressing hM4Di only and no cells that expressed ChR2-EYFP alone. The majority of labelled neurons were found in the PrL cortex (PrL vs medial orbital 72±1.5% vs 10.8±4.6%; *Figure 1D and E*). Successful targeting of the PrL-P neurons was confirmed by the presence of hM4Di-mCherry and Chr2-EYFP labelled fibres within the ventrolateral (vl)PAG (*Figure 1F*). In control animals, in which no CAV–CMV–CRE was delivered to the vlPAG, there was negligible expression of actuator protein in the mPFC after delivery of Cre-dependent AAV vectors (*Figure 1—figure supplement 1*).

### PrL-P neurons bidirectionally regulate nociception in naive rats

To determine the effect of PrL-P neurons on noxious withdrawal threshold in healthy animals, ChR2/hM4Di-expressing (Naive[PrL-P.ChR2:hM4Di]) and control (Naive[PrL-P.Control]) rats underwent Hargreaves' testing with opto-activation or chemo-inhibition of PrL-P neurons (*Figure 2A–F*). Opto-activation of PrL-P neurons (10–15 mW, 20 Hz, 10 ms pulse) in Naive[PrL-P.ChR2-hM4Di] rats produced a significant increase in thermal withdrawal latencies ipsilateral, but not contralateral, to the transfected PrL-P pathway (baseline vs PrL-P opto-activation=7.5±0.4 vs 10.4±0.9 s, p=0.008, paired t-test, n=10; ; *Figure 2B and C*). The equivalent illumination paradigm in Naive[PrL-P.Control] rats did not alter ipsilateral or contralateral withdrawal latencies (*Figure 2B, C and D*). Conversely, chemo-inhibition

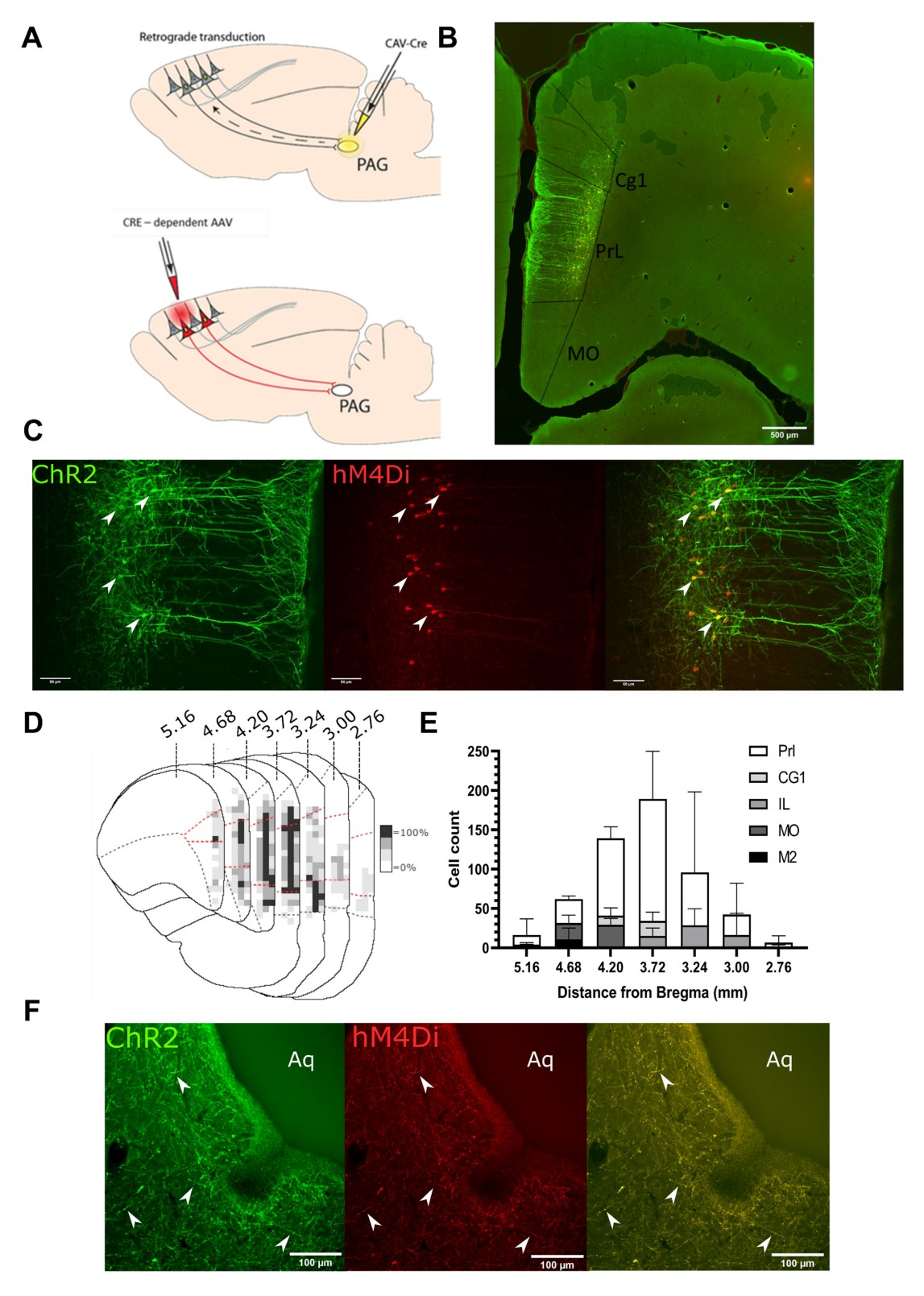

**Figure 1.** Transfected mPFC→PAG neurons arise mainly from the pre-limbic (PrL) cortex. (**A**) Intersectional viral vector strategy. We used a retrograde canine adenovirus and Cre-dependent adeno-associated viral vectors to express genetically encoded actuators (both channelrhodopsin-2 [ChR2] and hMD4i) within medial prefrontal cortex (mPFC) neurons that project to the periaqueductal grey (PAG). (**B**) Photomicrograph of mPFC showing labelled neurons residing mainly in the PrL cortex. (**C**) PrL cortex with colocalisation of mCherry (hM4Di) and EYFP (ChR2) in neurons projecting to PAG (many

*Figure 1 continued on next page*

*Figure 1 continued*

examples but several marked with white arrows). (**D**) Conjunction plot illustrating location of mPFC→PAG neurons throughout the mPFC (n=3 rats). Darker shading indicates positional overlap of positively labelled (hM4Di) neurons from more than one animal (light=1 animal, mid=2, and dark=3). Dotted red line demarks the PrL cortex. (**E**) Comparative distribution of mPFC→PAG neurons throughout the cortex (mean ± SEM). (**F**) Photomicrograph showing ChR2-EYFP and hM4Di-mCherry containing fibres from mPFC projecting to the ventrolateral region of PAG (many examples but several marked with white arrows).

The online version of this article includes the following figure supplement(s) for figure 1:

**Figure supplement 1.** There was negligible expression of channelrhodopsin-2 (ChR2)-EYP and hM4Di-mCherry in control animals that did not receive CAV–CMV–CRE into the periaqueductal grey.

(2.5 mg·kg$^{-1}$ CNO i.p.) of PrL-P neurons in the same Naive$^{PrL-P.ChR2-hM4Di}$ rats that received opto-activation significantly decreased the average withdrawal ipsilateral, but not contralateral, to the transfected PrL-P pathway. (Baseline vs chemo-inhibition of PrL-P=10.3±0.6 vs 8.3±0.6s, p=0.006, paired t-test, n=15; *Figure 2E and F*.) CNO had no significant effect on withdrawal thresholds in Naive$^{PrL-P.ChR2-hM4Di}$ rats (*Figure 2E and F*). These findings demonstrate that there is a tonic level of activity within the PrL-P pathway that dynamically regulates nociception in the absence of any process of sensitization.

## Tonic activity in PrL-P neurons delays the development of neuropathic hypersensitivity

The tibial nerve transection (TNT) model of neuropathic pain was used to assess the contribution of PrL-P neurons to the development of sensitisation (*Figure 3A–D*). TNT$^{PrL-P.ChR2-hM4Di}$ and TNT$^{PrL-P.Control}$ rats had nociceptive sensory testing before and after CNO (2.5 mg·kg$^{-1}$ i.p., *Figure 3C*) longitudinally up to 42 days post-nerve injury (*Figure 3—figure supplement 1*). Chemo-inhibition of PrL-P neurons unmasked mechanical and cold hypersensitivity in TNT$^{PrL-P.ChR2-h M4Di}$ rats, for the ipsilateral, nerve-injured, hindpaw at day 3 post-nerve injury (*Figure 3E and I*). The mechanical withdrawal threshold (von Frey [vF]) was reduced on average by 80% on day 3 post-TNT, from 6.0±1.3 g (pre-CNO) to 1.2±0.5 g (post-CNO) (two-way ANOVA, CNO F(1,30)=20.09, p=0.0001; Sidak's post-test day 3 pre-CNO vs post-CNO, p=0.008, n=16; *Figure 3E*). Similarly, the number of cold-evoked nocicifensive behaviours (foot flicking, biting, and grooming) was significantly increased ipsilaterally by PrL-P chemo-inhibition at day 3 post-TNT from 2.8±0.5 to 5.8±0.7 events (two-way ANOVA, CNO F(1,30)=9.6, p=0.004; Sidak's post-test day 3 pre-CNO vs post-CNO, p=0.003, n=16; *Figure 3I*). At 7 days post-nerve injury, PrL-P chemo-inhibition also significantly decreased the ipsilateral mechanical withdrawal threshold from 2.5±0.5 to 0.30±0.06 g (two-way ANOVA, CNO F(1,30)=20.09, p=0.0001; Sidak's post-test p=0.001, n = 16; *Figure 3E*). For the contralateral (uninjured) paw, PrL-P chemo-inhibition significantly reduced mechanical withdrawal thresholds at day 3 post-TNT from 13.5±0.7 to 9.0±1.4 g (two-way ANOVA, CNO F(1,30)=5.77, p=0.02; Sidak's post-test day 3 pre-CNO vs post-CNO, p=0.03, n=16) but not thereafter (*Figure 3F*). From 14 days post-nerve injury and up to 42 days, PrL-P chemo-inhibition ceased to significantly change either mechanical or cold-evoked nocifensive behaviour on the ipsilateral hindpaw (*Figure 3E and I*, *Figure 3—figure supplement 1*). In TNT$^{PrL-P.Control}$ rats, CNO failed to significantly change either mechanical withdrawal thresholds or cold (acetone)-evoked nocicfensive behaviour on either the ipsilateral or contralateral paw at any timepoint post-TNT (*Figure 3G,H, I, and K*). Additonally, an equivalent vehicle injection delivered at 7 days post-nerve injury did not affect pain-like behaviour in TNT$^{PrL-P.ChR2-hM4Di}$ rats (*Figure 3—figure supplement 2*) These results suggest that PrL-P neurons provide a tonic descending drive to oppose peripheral sensitisation during the early stages of the development of neuropathic pain, but this effect is lost as sensitisation becomes established after 14 days.

## Chemogenetic inhibition of PrL-P neuronal activity is aversive in TNT rats with latent sensitisation

Neuropathic sensitisation is associated with negative affect (*King et al., 2009*; *Hirschberg et al., 2017*), which raises the possibility that PrL-P neurons act to oppose the development of negative affect. If so, then chemo-inhibition of PrL-P neurons in the early phase after nerve injury would be

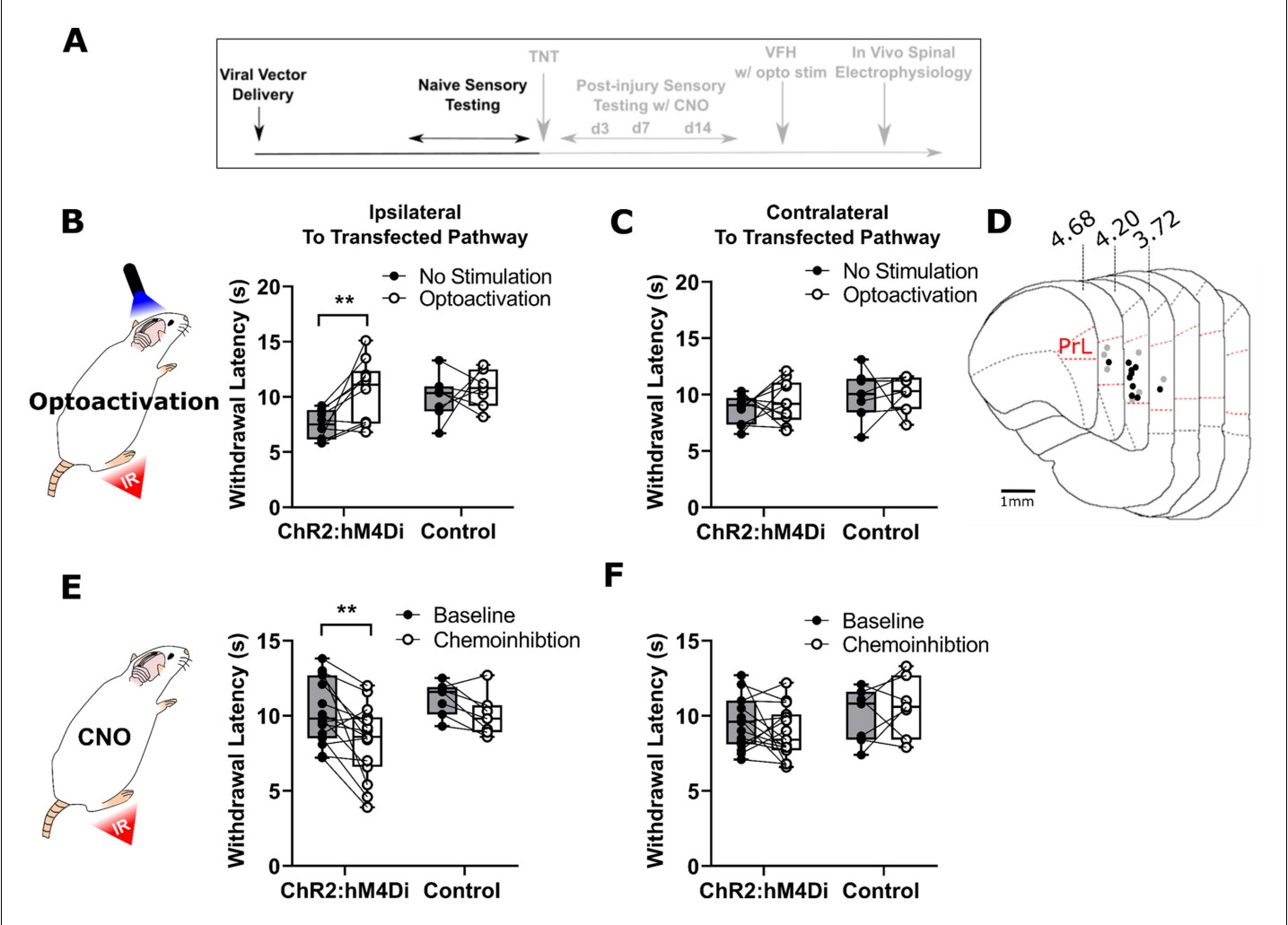

**Figure 2.** PrL-P neurons bidirectionally regulate nociception in naive rats. (A) Experimental timeline. (B) Illumination of PrL (445 nm, 20 Hz, 10–15 mW, 10 ms pulse, concomitant with hind-paw heating) in Naive[PrL.ChR2:hM4Di] rats increased thermal withdrawal latencies of the ipsilateral hindpaw but not in Naive[PrL.Control] rats that did not express channelrhodopsin-2 (ChR2; paired t-test, t(9)=3.37, p=0.008, n=10 for ChR2:hM4Di group; t(6)=0.63, p=0.55, n=7 for control group). (C) Equivalent illumination of PrL had no effect on the thermal withdrawal latency of the contralateral hindpaw in either Naive[PrL.ChR2:hM4Di] or Naive[PrL.Control] rats (paired t-test, t(9)=0.86, p=0.40, n=10 for Naive[PrL.ChR2:hM4Di] rats; t(6)=0.14, p=0.90, n=7 for Naive[PrL.Control] rats). (D) Optic fibre tip locations in the medial prefrontal cortex from Naive[PrL.ChR2:hM4Di] (•) and Naive[PrL.Control] (O) rats. For simplicity, fibre placements are depicted in a single hemisphere. (E) Systemic CNO (2.5 mg·kg⁻¹ i.p.) in Naive[PrL.ChR2:hM4Di] rats decreased withdrawal latencies (mean value at 20–40 min post-injection) of the ipsilateral paw but not in Naive[PrL.Control] rats (paired t-test, t(14)=3.26, p=0.006, n=15 for Naive[PrL.ChR2:hM4Di] rats; t(6)=0.63, p=0.55, n=7 for Naive[PrL.Control] rats). (F) CNO had no effect on the thermal withdrawal latency of the contralateral hindpaw in either Naive[PrL.ChR2:hM4Di] rats or Naive[PrL.Control] rats (paired t-test, t(14)=1.22, p=0.24, n=15 for Naive[PrL.ChR2:hM4Di] rats; t(6)=0.43 p=0.68, n=7 for Naive[PrL.Control] rats).

The online version of this article includes the following source data for figure 2:

**Source data 1.** Numerical data to support graphs in *Figure 2*.

expected to cause aversion. To test this proposition, TNT[PrL-P.ChR2-hM4Di] and TNT[PrL-P.Control] rats had place aversion testing with CNO conditioning between days 2 and 5 post-TNT (*Figure 4A*). TNT[PrL-P.ChR2-hM4Di] animals showed an aversion to the CNO paired chamber (post-conditioning–pre-conditioning time=−82.9±24.7 s, n=8; *Figure 4B and C*). We calculated the preference of each rat for the CNO or vehicle paired chamber and found TNT[PrL-P.ChR2-hM4Di] animals showed a significantly reduced preference score compared to the vehicle paired chamber (*Figure 4C*; CNO paired vs vehicle paired=0.8±0.04 vs 1.06±0.06, paired t-test, p=0.04, n = 8). TNT[PrL-P.Control] animals showed no difference in preference score for CNO and vehicle paired chambers (*Figure 4D*, CNO paired vs vehicle paired=1.1±0.18 vs 1.00±0.21, paired t-test, p=0.81, n=9). These findings are consistent with

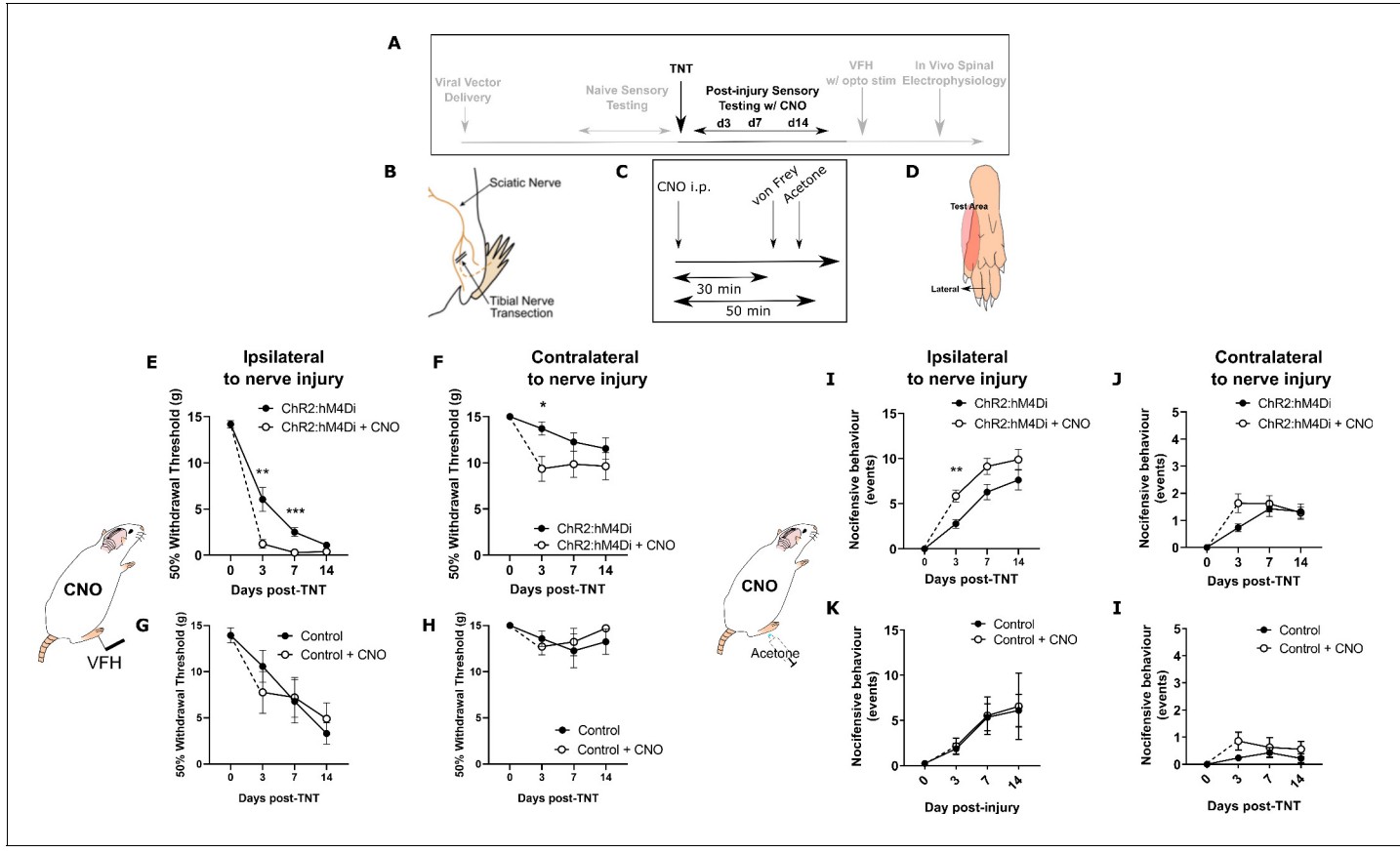

**Figure 3.** Inhibition of PrL-P neurons unmasks hypersensitivity in neuropathic rats. (A) Experimental timeline. (B) Tibial nerve transection (TNT) was used to produce the neuropathic injury. (C)- Sensory testing was conducted at 30 min after systemic delivery of CNO and (D) testing was conducted on the lateral plantar surface of the hindpaw in a receptive field adjacent to injured tibial nerve. (E) In TNT$^{PrL.ChR2:hM4Di}$ rats, CNO (2.5 mg·kg$^{-1}$ i.p.) reduced the mechanical withdrawal threshold at 3 and 7 days post nerve injury on the ipsilateral (injured) hindpaw (two-way ANOVA, main effect CNO, F(1,30)=20.09, p=0.0001; timexCNO, F(2, 60)=6.892, p=0.002; Sidak's post-test day 3, p=0.008; day 7, p=0.001, n = 16) and (F) on the contralateral paw at 3 days post-injury (two-way ANOVA, CNO F(1,30)=5.77, p=0.02; Sidak's post-test, p=0.02, n = 16). (G and H) In TNT$^{PrL.Control}$ rats, the same dose of CNO did not alter mechanical withdrawal thresholds on either the ipsilateral or contralateral hindpaw (two-way ANOVA, main effect; ipsilateral CNO, F (1,14)=0.02, p=0.90, n=8 and contralateral CNO, F(1,14)=0.15, p=0.71, n=8, respectively). (I and J) In TNT$^{PrL.ChR2:hM4Di}$ rats, CNO increased acetone-evoked nocifensive events at 3 days post-injury on the ipsilateral paw (two-way ANOVA, main effect CNO, F(1,30)=9.6, p=0.004; Sidak's post-test, p=0.003, n=16) but not contralaterally (two-way ANOVA, main effect CNO, F(1,29)=1.3, p=0.26, n=16). (K and I) In TNT$^{PrL.Control}$ rats, CNO did not alter acetone-evoked nocicfensive behaviour (two-way ANOVA, main effect CNO ipsilateral, F(1,12)=0.02, p=0.89, n=7 and main effect CNO contralateral, F (1,12)=2.2, p=0.16, n=7).

The online version of this article includes the following source data and figure supplement(s) for figure 3:

**Source data 1.** Numerical data to support graphs in *Figure 3*.

**Figure supplement 1.** Chemo-inhibition of PrL-P neurons affects nocicfensive behaviour in early but not late timepoints post-injury in neuropathic animals.

**Figure supplement 1—source data 1.** Numerical data to support graphs in *Figure 3—figure supplement 1*.

**Figure supplement 2.** Delivery of vehicle does not affect sensitisation in TNT$^{PrL.ChR2:hM4Di}$ rats at 7 days post-TNT.

**Figure supplement 2—source data 1.** Numerical data to support graphs in *Figure 3—figure supplement 2*.

PrL-P neuronal activity opposing the development of negative affect in the immediate period after nerve injury.

## Restoration of PrL-P neuronal tone attenuates allodynia in established neuropathic sensitisation

To test whether the loss of function by PrL-P neurons in later stage neuropathic sensitisation could be reversed, we employed opto-activation of PrL-P neurons to test if it was still able to suppress sensitisation (*Figure 5*). Opto-activation in TNT$^{PrL-P.ChR2:hM4Di}$ rats (20 Hz, 10 ms, 10–15 mW) produced

an increase in the mechanical withdrawal threshold (baseline vs opto-activation vs recovery=1.7±0.5 vs 5.2±1.4 vs 2.1±0.5 g, one-way repeated-measures (RM) ANOVA, p=0.02; Sidak's post-test baseline vs opto-activation, p=0.01, n=9; *Figure 5B*). Equivalent illumination in TNT^PrL-P.control rats did not change the mechanical withdrawal threshold (baseline vs opto-activation vs recovery=1.4±0.5 vs 1.0±0.4 vs 1.4±0.3, one-way RM ANOVA, p=0.61, n=3; *Figure 5B*). This data indicates that the PrL-P neurons are still capable of supressing neuropathic sensitization in late-stage TNT rats.

## PrL-P produces antinociception in neuropathic pain by inhibition of dorsal horn nociceptive responses

To better understand the mechanism by which the PrL-P neurons suppress neuropathic sensitisation, TNT^PrL-P.ChR2:hM4Di rats were tested in acute spinal electrophysiology experiments. Opto-activation of PrL-P neurons attenuated the evoked responses of spinal dorsal horn wide dynamic range (WDR) neurons (*Figure 5C–F*). The number of action potentials evoked by a punctate mechanical stimulus with a 4 and 15 g vF hair was reduced on average by 43 and 23%, respectively (*Figure 5D and E*; 4 g vF, baseline vs opto-activation vs recovery=25.7±5.0 vs 14.63±3.1 vs 25.0±45.0 action potentials; mixed model [REML], fixed effect treatment, p=0.007; Dunnett's post-test baseline vs opto-activation, p=0.009, n=9; *Figure 5E*; 15 g vF, baseline vs opto-activation vs recovery=45.17±6.9 vs 34.5±6.9 vs 43.9±7.8; mixed model [REML], p=0.10, Dunnett's post-test baseline vs opto-activation, p=0.04, n=9). Similarly, cold-evoked spinal WDR neuron activity was significantly reduced by opto-activation of PrL-P neurons (*Figure 5F*, average reduction of 47%, baseline vs opto-activation=172.0±38.8 vs 91.7±42.2 action potentials, paired t-test, p=0.04, n=4). This data indicates that the PrL-P neurons are acting to suppress neuropathic sensitisation (punctate and cold allodynia) at a spinal level through the engagement of the DPMS.

## Discussion

By using a longitudinal study design, we have been able to reveal the dynamic contributions of PrL to PAG communication to neuropathic pain state development. In uninjured animals, we found PrL-P neurons to exert tonic inhibitory control over evoked noxious withdrawal responses, suggesting they were involved with the moment-to-moment regulation of nocifensive behaviour. Following tibial nerve injury, rats developed mechanical and thermal allodynia that plateaued at around 14 days post injury. Chemo-inhibition of PrL-P neurons at 3 and 7 days post-injury revealed latent hypersensitivity both ipsilateral and contralateral (day 3 only) to nerve injury and produced place aversion in a conditioned place preference paradigm. However, chemo-inhibition of PrL-P neurons at more than 14 days post-injury failed to significantly affect mechanically or thermally evoked pain-like behaviour. These findings are consistent with there being a tonic activity in PrL-P neurons that suppresses hypersensitivity early after nerve ligation, but this is lost with time as the neuropathic pain phenotype emerges. Despite the loss in function, opto-activation of PrL-P neurons during later stages of neuropathic pain produces anti-allodynic effects in neuropathic animals achieved, at least in part, by inhibitory effects on spinal nociceptive processing. We postulate that this cortical–midbrain–spinal network allows executive control of nociception and regulation of pain, and that it is the loss of cortical drive to the DPMS, which is a major contributor to the expression of neuropathic sensitisation after nerve injury.

Loss of endogenous inhibitory control of CNS pain processing (*Staud, 2012*) and altered functional connectivity between the mPFC and the PAG is a shared feature of a wide variety of human chronic pain conditions (*Jensen et al., 2012*; *Yu et al., 2014*; *Chen et al., 2017*; *Segerdahl et al., 2018*)—our findings suggest these changes are causally related. The human mPFC, including the anterior cingulate cortex (ACC), is increasingly recognised as a key locus in the development and maintenance of chronic pain, with changes in structure and function that are associated with, and sometimes predictive of, the transition to chronic pain (*Apkarian et al., 2004*; *Baliki et al., 2012*; *Hashmi et al., 2013*; *Baliki and Apkarian, 2015*). The PAG is not only a key node in the DPMS but also an important orchestrator of autonomic and sensorimotor systems that is engaged to support mPFC function in aversive learning, emotional modulation, and pain modulation, which are all relevant to the chronic pain phenotype (*Keay and Bandler, 2001*; *Franklin et al., 2017*; *Rozeske et al., 2018*; *Huang et al., 2019*). In humans, changes in functional connectivity between the mPFC/ACC and PAG are commonly observed in experimental paradigms that produce emotional,

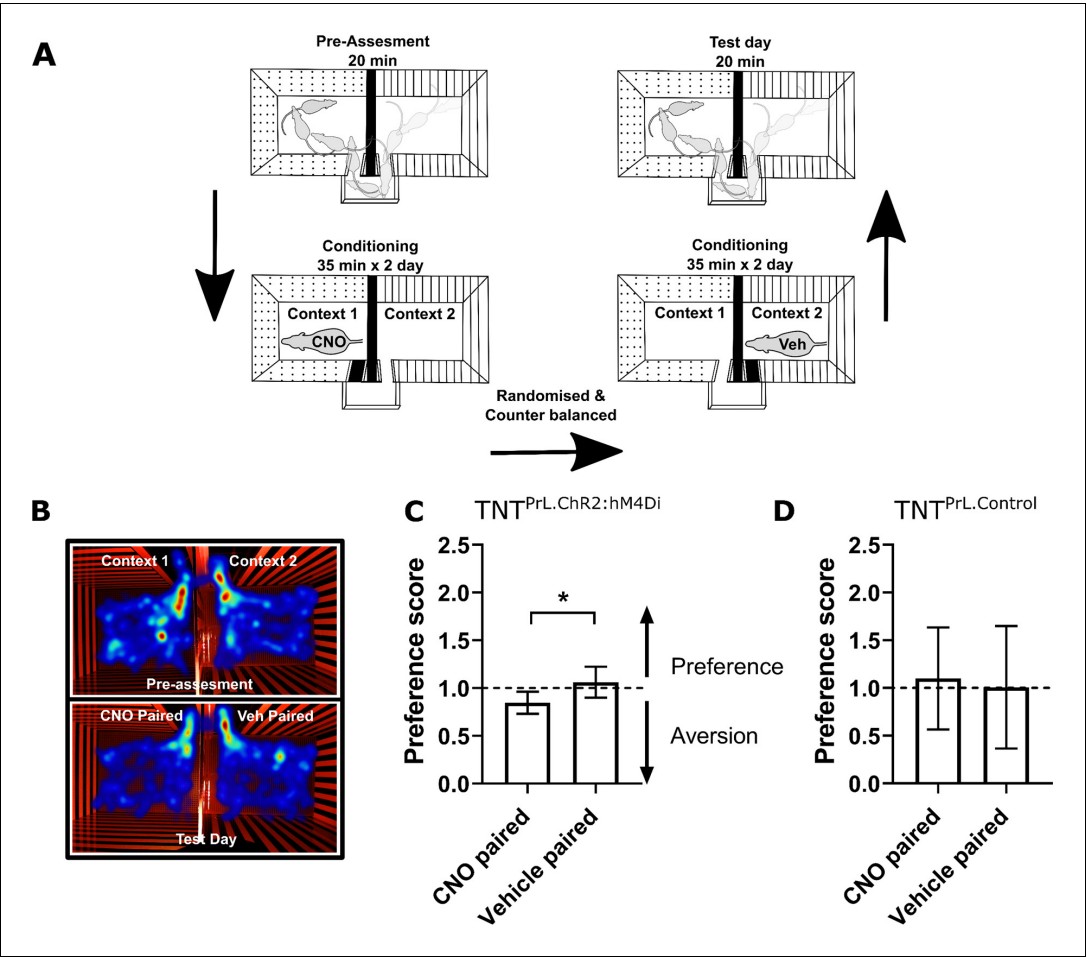

**Figure 4.** Inhibition of PrL-P neurons produces aversion in neuropathic animals. (**A**) Conditioned place aversion protocol. (**B**) Example heatmap visualisation of the time spent within the testing chambers prior (top) and following conditioning with CNO or vehicle. (**C**) Group data showing conditioning with CNO in TNT[PrL.ChR2:hM4Di] rats 2–5 days after tibial nerve transection (TNT) produced place aversion (paired t-test, t(7)=2.43, p=0.04, n=8). (**D**) CNO administration to TNT[PrL.Control] rats did not produce place aversion (paired t-test, t(8)=0.25, p=0.81, n=9). The online version of this article includes the following source data for figure 4:

**Source data 1.** Numerical data to support graphs in *Figure 4*.

attentional, and placebo/nocebo influences on pain as well following the delivery of analgesic drugs and often interpreted as reflecting engagement of the DPMS (*Wager et al., 2004*; *Wiech et al., 2014*; *Wanigasekera et al., 2018*; *Oliva et al., 2021*). Moreover, changes in the functional connectivity between regions of the mPFC and the PAG are often correlated with changes in pain perception and/or disease progression (*Cifre et al., 2012*; *Hemington and Coulombe, 2015*; *Harper et al., 2018*; *Segerdahl et al., 2018*; *Wanigasekera et al., 2018*; *González-Roldán et al., 2020*). Here, we provide evidence in rodents that the PrL, a component of the rodent mPFC, can engage the DPMS to affect nociception, and loss in PrL-P neuron function is causally related to the development of the neuropathic pain state in rats. We suggest that changes in functional communication between the mPFC and PAG, whether trait-, age- or disease-related, likely manifest as alterations in the descending control of spinal nociception.

Preclinical findings suggest that loss of mPFC–PAG functional communication is explained by both local and inter-regional network alterations. In neuropathic rodents, at 7–10 days post-nerve injury, there is a decline in spontaneous and evoked PrL layer five pyramidal cell activity including those that project to the PAG (*Cheriyan and Sheets, 2018*; *Mitrić et al., 2019*). This reduction in PrL projection neurons excitability is produced in part by enhanced feedforward inhibition from local

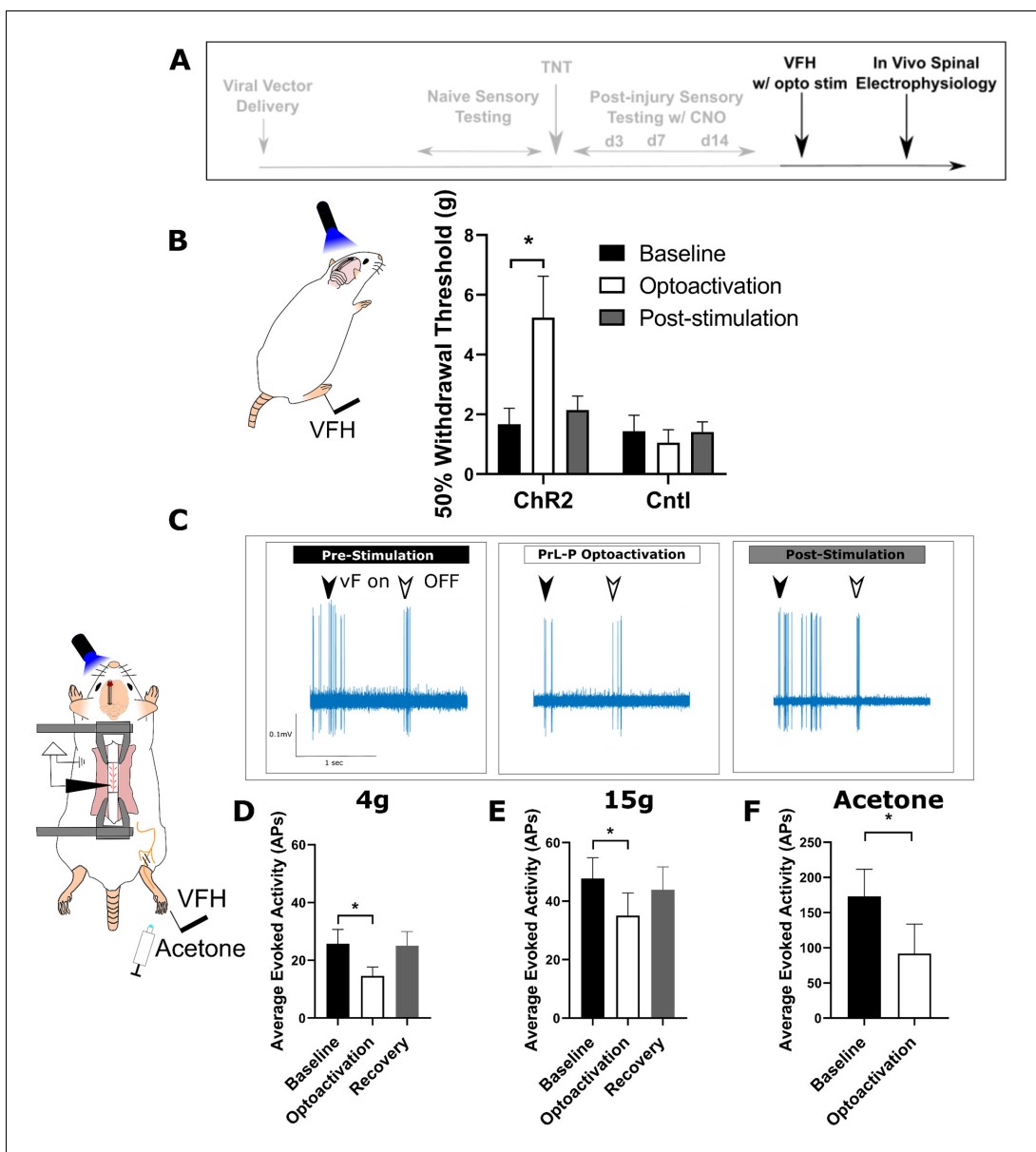

**Figure 5.** Activation of PrL-P neurons produces antinociception in neuropathic rats by inhibition at a spinal level. (A) Experimental timeline. (B) Delivery of blue light (445 nm, 20 Hz, 10–15 mW, 10 ms pulse, concomitant with hind-paw stimulation) produced a significant increase in mechanical withdrawal threshold of the injured (ipsilateral) hind-paw in TNT[PrL.ChR2:hM4Di] rats (repeated-measures [RM] ANOVA, treatment, F(1.12, 8.96)=8.07, p=0.02; Sidak's post-test, *p<0.05, n=9) but not in TNT[PrL.Control]rats (RM ANOVA, treatment, F(1, 2)=0.35, p=0.61, n=3). (C) Example raw data trace illustrating suppression of von Frey hair evoked spinal dorsal horn neuron activity during blue light (420 nm, 10–15 mW, 10 ms duration, concomitant with hind-paw stimulation) delivery to the PrL in TNT[PrL.ChR2:hM4Di] rats (arrows demark beginning and end of stimulus). (D) Group data illustrating suppression of 4 g evoked spinal dorsal horn neuronal activity by illumination of the PrL in TNT[PrL.ChR2:hM4Di] rats (mixed model [REML], fixed effect opto-activation, F[1.99, 13.98]=7.18, p=0.007; Dunnet's post-test baseline vs opto-activation, p=0.009, n=10). (E) Illumination of the PrL in TNT[PrL.ChR2:hM4Di] rats also supressed 15 g evoked spinal dorsal horn neuronal activity (mixed model [REML], fixed effect opto-activation, F(1.52, 10.64)=2.94, p=0.10, Dunnet's post-test baseline vs opto-activation, p=0.046, n=10.) (F) Delivery of blue light to the Prl in TNT[PrL.ChR2:hM4Di] rats decreased acetone-evoked spinal dorsal horn neuronal activity (paired t-test, t(3)=3.58, p=0.04, n=4).

The online version of this article includes the following source data for figure 5:

**Source data 1.** Numerical data to support graphs in *Figure 5*.

GABAergic interneurons and driven by inputs from the basolateral amygdala (*Zhang et al., 2015*; *Cheriyan et al., 2016*; *Kiritoshi et al., 2016*; *Cheriyan and Sheets, 2018*; *Huang et al., 2019*). Opto-inhibition of BLA inputs to the PrL releases PrL-P neurons to engage descending inhibitory control from the PAG (*Huang et al., 2019*). PrL-P neurons are glutamatergic and target both GABAergic and glutamatergic neurons in the PAG (*Franklin et al., 2017*; *Huang et al., 2019*) but engage descending inhibitory control from the PAG that is associated with the release of local GABAergic control (*Tovote et al., 2016*; *Huang et al., 2019*). We suggest that the BLA–PrL–PAG–spinal network is central to the expression of neuropathic pain and therapeutics that reengage cortical control of descending pain modulation may be effective in treating both sensory and affective disturbances in chronic pain. However, while BLA inputs to the PrL drive sensory hypersensitivity and negative affect by reducing descending inhibition of the spinal dorsal horn, BLA inputs to the dorsal cingulate regions mitigate pain-related aversion (*Meda et al., 2019*). Additionally, in neuropathic rodents, the E/I ratio of BLA inputs into infralimbic projections to the PAG remain unchanged (*Cheriyan and Sheets, 2018*). Thus, there appears region-specific alterations to BLA–mPFC–PAG neuronal network that must be considered if novel and effective CNS therapeutic strategies are to be realized.

We found that PrL-P neurons alter sensory and affective aspects of neuropathic pain, at least in part, via actions on spinal dorsal horn nociceptive processing. Recently, *Huang et al., 2019* dissected BLA–PrL–PAG circuitry and demonstrated contributions of spinal noradrenergic alpha-2 adrenoceptor and 5-hydroxytryptamine receptor 1/2 signalling to PrL effects on pain-like behaviour in neuropathic rats; *Huang et al., 2019*. However, the expression of these receptors in the spinal ventral horn confounds interpretation of effects on sensory/nociceptive versus motor processing (*Shi et al., 1999*; *Perrier et al., 2013*). Here we show that, in neuropathic animals, peripherally evoked spinal dorsal horn WDR neurons are inhibited by PrL-P neuronal activity confirming that the PrL cortex is able to engage DPMS that originate in the PAG. Spinal WDR neurons are a known target of descending modulatory systems and their activity correlates well with both withdrawal reflexes and pain perception (*Maixner et al., 1986*; *You et al., 2003*; *McMullan and Lumb, 2006a*; *Drake et al., 2016*). It is significant from a therapeutic perspective that the PrL is able to affect nociceptive information early in the ascending pathway, likely prior to extensive integration with other nociceptive/non-nociceptive information, which would allow for selective and potent actions on the pain experience and pain state development (*Heinricher et al., 2009*).

Spinal noradrenergic alpha2adrenoceptor signalling masks latent mechanical and cold allodynia at early, but not late, timepoints post-tibial nerve injury and shows a remarkably similar chronology to that observed in this study (*Hughes et al., 2013*). Notably, in addition to affecting the sensitivity of the injured paw, antagonism of spinal alpha2-adrenoceptors reveals contralateral allodynia (*Hughes et al., 2013*), something also observed after chemo-inhibition of PrL-P neurons. This indicates that nerve injury drives central sensitisation that affects the response to nociceptive inputs from the uninjured paw to produce contralateral hypersensitivity that can be revealed by blockade of the descending modulation of spinal processing originating from the PrL cortex.

These observations, supported by those of Huang et al, suggest that spinal noradrenaline (NA) release mediates a large part of the PrL analgesic actions in the spinal dorsal horn (*Hughes et al., 2013*; *Hughes et al., 2015*; *Huang et al., 2019*). Our findings indicate that the progressive loss of spinal noradrenergic tone in this neuropathic model is due to the loss in top-down executive control from PrL-P neurons, but this is yet to be definitively tested. Despite this loss in function, potentiation of spinal NA signalling with reuptake inhibitors can prevent the development of neuropathic pain and reverse neuropathic hypersensitivity in late-stage neuropathic animals (*Hughes et al., 2015*), and NA reuptake inhibitors are currently used to treat neuropathic pain in human patients (*Finnerup et al., 2015*). Therapeutic strategies aimed at lifting the enhanced feedforward inhibition in the PrL combined with potentiation of spinal NA signalling could provide a novel approach to treat neuropathic pain in humans. However, targeted approaches are likely necessary as chemo-activation of locus coeruleus projections to the spinal dorsal horn and mPFC produce pain relief and conflicting negative affect in neuropathic rats (*Hirschberg et al., 2017*).

At early post-injury timepoints, chemo-inhibition of PrL-P neurons produced place aversion indicating that loss of cortical control over the PAG and associated DPMS worsens the affective state of injured animals. This change in affective state is likely secondary to effects on spinal nociceptive processing as neuropathic animals show ongoing pain-like behaviour and its relief, for instance by

stimulating spinal NA receptors, produces place preference in neuropathic rats (*King et al., 2009*; *Hirschberg et al., 2017*). However, direct effects of PrL-P neurons on affective processing should not be overlooked given the role of the PAG in fear, anxiety, and depression (*Tovote et al., 2015*; *George et al., 2019*). Interestingly, *Rozeske et al., 2018* showed dorsal mPFC projections to the PAG regulate the appropriate expression of aversive memories suggesting that loss of functional communication between the PrL and PAG may lead to generalised aversion and negative affect; *Rozeske et al., 2018*. The PrL has previously been shown to contribute to pain-related anxiety, deficiencies in reward processing, and avoidance behaviour (*Lee et al., 2015*; *Wang et al., 2015*; *Zhang et al., 2015*; *Liang et al., 2020*). Our findings add to the growing picture of mPFC-PAG communication as a critical regulator in a range of disease-relevant features including emotional coping, aversive learning, and autonomic control including nociceptive processing and, now, neuropathic pain state development (*Franklin et al., 2017*; *Rozeske et al., 2018*; *Huang et al., 2019*).

A 200-μm tapered optic fibre was inserted into the PrL to allow for light delivery in opto-activation experiments. This likely led to some damage/disruption to overlying cortical tissue in the cingulate cortex area 1 (CG1) area that is commonly referred to as the rodent ACC. We consider the impact of these on animal behaviour to be minimal as tapered fibres produced negatable tissue damage in post-mortem histological sections. However, a consideration of any 'off-target' effects is presented for the reader. Lesion or pharmacological inactivation of the equivalent CG1 region has been shown to not affect sensory thresholds and/or pain-like behaviour in rodents (*Johansen et al., 2001*; *Johansen and Fields, 2004*; *Navratilova et al., 2015*); therefore, we are confident that effects observed in our chemo-inhibition experiments are specific for CNO. Furthermore, the CG1/ACC region has been shown to generate change in affective state to guide adaptive behavioural and learning, particularly place avoidance (*Johansen et al., 2001*; *Johansen and Fields, 2004*). Our data demonstrate that rats implanted with tapered optic fibres are still able to undergo place conditioning, indicating that any cortical disruption produces by our fibres does not impact CG1/ACC function relevant to our investigation.

In summary, we have identified specific contributions of PrL-P neurons to regulate nociception in healthy animals and charted their dynamic contributions to neuropathic pain state development following injury. Our findings suggest that PrL-P neurons engage descending inhibitory control of the spinal dorsal horn to regulate CNS nociceptive processing and moment-to-moment noxious threshold to affect behaviour. Following nerve injury, tone in PrL-P neurons initially constrains the spatio-temporal development of neuropathic hypersensitivity, but there is a progressive loss in the functional communication between the PrL and PAG as the pain state develops. Our findings aid interpretation of human clinical observations that demonstrate altered functional connectivity between the mPFC and PAG is an important mechanism in the development of chronic pain.

## Materials and methods

### Key resources table

| Reagent type (species) or resource | Designation | Source or reference | Identifiers | Additional information |
|---|---|---|---|---|
| Strain, strain background | Wistar (rat) male | Envigo, NL | RCCHan | |
| Transfected construct | Canine adenoviral vector | IGMM, FR | CAV2-CMV-CRE | |
| Transfected construct | Adeno-associated viral vector | UNC Vector Core, USA | AAV2-EF1a-DIO-hChR2-EYFP | |
| Transfected construct | Adeno-associated viral vector | Addgene, USA | AAV2-hSyn-DIO-hM4Di-mCherry | |
| Chemical compound | Clozepine-N-oxide | Tocris Bioscience, UK | 4936 | |
| Antibody | Anti-GFP (chicken polyclonal) | Abcam, USA | Ab13970 | (1:5000) |

*Continued on next page*

*Continued*

| Reagent type (species) or resource | Designation | Source or reference | Identifiers | Additional information |
|---|---|---|---|---|
| Antibody | Anti-mCherry (rabbit polyclonal) | BioVision | 5993 | (1:2000) |
| Other | Optic fibre | Optogenix, IT | Lambda-B | NA 0.66, length 4.4 mm, and light-emitting length 2 mm |
| Other | Q-probes | Neuronexus, USA | Q1 × 1-tet-10mm-121-Q4 | |
| Other | Headstage chip | Intan Technologies, USA | RHD2132 | |
| Other | Diode laser | Omricon Laserage, DE | LuxX445-100 | 445 nm/100 mW |
| Other | OpenEphys Acquisition System | OpenEphys, USA | | |
| Software, algorithm | OpenEphys GUI | OpenEphys, USA | | |
| Software, algorithm | EthovisionXT | Noldus, NL | | |
| Software, algorithm | Prism | GraphPad, USA | | |
| Software, algorithm | GNU Image Manipulation Program | GIMP, USA | | |
| Software, algorithm | BORIS event logging software | BORIS, IT | | |

## Animals

All experimental and surgical procedures were conducted in accordance with the UK Animals (Scientific Procedures) Act (1998) and local ethical review. Adult male Wistar rats (n=56, 250–275 g; Envigo, NL) were housed in the University of Bristol's Animal Services Unit with cage enrichment (e.g. cardboard tubes and wooden chews) on a reversed light cycle and with food/water provided ad libitum. Where possible, animals were group-housed but were singly housed for up to 7 days while healing from surgery occurred.

## Experimental design

This study's primary objective was to investigate the contribution of PrL-P neurons to the development of sensory and affective aspects of neuropathic pain. To achieve this, opto- and chemogenic actuator proteins were expressed in PrL-P neurons to enable interrogation of the behavioural and neurophysiological consequences of their selective and specific opto-activation and chemo-inhibition. To selectively express actuator proteins in only PrL-P afferents, we used an intersectional, Cre-dependent viral vector approach (*Boender et al., 2014*). Twelve animals were used to develop the intersectional viral vector methodology in vivo. Briefly, Cre-dependent adenoviral vectors encoding ChR2 or the inhibitory DREADD, hM4Di, were delivered to the PrL. To restrict their expression to only those PrL neurons that project to the PAG, we delivered a retrograde canine adenoviral vector (CAV2) that encodes Cre-recombinase to the PAG (*Hnasko et al., 2006*). CAV2 gains access to neurons primarily via their synaptic terminals (*Soudais et al., 2001*) before being transported retrogradely to the neuronal cell body leading to Cre expression. Thus, Cre-dependent expression of actuator proteins will only occur in those PrL neurons that synapse in the PAG. Control animals had injection of Cre-dependent vectors to the cortex but no CAV-CMV-CRE to the PAG. Without Cre, there should be no expression of actuator proteins allowing the evaluation of off-target effects of CNO (as well as identification of nonspecific expression of actuators). Following the expression of actuator proteins, the effect of selective opto-activation and chemo-inhibition of PrL-P neurons on sensory (n=24) and affective aspects (n=20) of pain-like behaviour was assessed in neuropathic (TNT[PrL-P.ChR2-hM4Di] and TNT[PrL-P.Control]) and uninjured (Naive[PrL-P.ChR2-hM4Di] and Naive[PrL-P.Control]) rats. This investigation used a longitudinal design in which the contribution of PrL-P neurons to pain-like behaviour and nociceptive processing were assessed before and up to 42 days following peripheral nerve injury. Five neuropathic rats were then used in acute spinal electrophysiological experiments to assess the effects of PrL-P neurons on nociceptive processing in the spinal dorsal horn.

Animals were assigned to experimental groups from different cages and selected at random but with no explicit randomisation protocol. The experimenter was blinded to the experimental group and test substance. Some animals were removed from the analysed data sets due to the following:

- Lack of or off-target transfection (n=4/44)
- Incorrect placement of optic fibres outside the PrL (n=3/24)
- Overt stress behaviour noted during experimental testing (n=3/44)
- Incompatible laser stimulation parameters (1/24)
- Incorrect dosing schedule in conditioned place aversion paradigm (1/20)

Where appropriate, removal from one experimental protocol did not mean removal from the entire investigation, as for example, incorrect placement of an optic fibre did not preclude data from this rat being included for chemo-inhibition experiments, which were not dependent on correct fibre placement.

Final experimental group numbers were as follows:

| Figure | Experiment | Rats (n) |
| --- | --- | --- |
| 2 | Opto-activation of PrL-P neurons in uninjured rats for Plantar Test | 10 Naive$^{PrL-P.ChR2-hM4Di}$ <br> 7 Naive$^{PrL-P.Control}$ |
| | Chemo-inhibition of PrL-P neurons in uninjured rats for Plantar Test | 16 Naive$^{PrL-P.ChR2-hM4Di}$ <br> 7 Naive$^{PrL-P.Control}$ |
| 3 | Chemo-inhibition of PrL-P neurons in tibial nerve transection (TNT) rats for sensory testing | 16 TNT$^{PrL-P.ChR2-hM4Di}$ <br> 7 TNT$^{PrL-P.Control}$ |
| 4 | Chemo-inhibition of PrL-P neurons in TNT rats for place aversion | 8 TNT$^{PrL-P.ChR2-hM4Di}$ <br> 9 TNT$^{PrL-P.Control}$ |
| 5 | Opto-activation of PrL-P neurons in TNT rats for sensory testing | 9 TNT$^{PrL-P.ChR2-hM4Di}$ <br> 3 TNT$^{PrL-P.Control}$ |
| | Opto-activation of PrL-P neurons for acute spinal electrophysiology | 5 TNT$^{PrL-P.ChR2-hM4Di}$ |

## Surgery

All surgeries were conducted using sterile technique. Throughout procedures, animals were kept hydrated and maintained at 37°C using a thermostatically controlled heat mat. Post-surgery, all animals were monitored closely until wounds had healed and the animal reached pre-surgery body weight.

## Stereotaxic injection/implants

Animals underwent recovery surgery for the delivery of viral vectors to the PAG and mPFC, and implantation of optic fibres over the PrL. Rats were anesthetised with Ketamine (50 mg·kg$^{-1}$; Zoetis, UK)/Medetomidine (0.3 mg·kg$^{-1}$; Vetoquinol, UK), prepared for surgery, and placed in a stereotaxic frame (Kopff, Germany). PrL-P projections extend bilaterally from each hemisphere with the ipsilateral projection being denser than the contralateral projection (~60 vs 40% of total labelled cells from retrograde tracing; *Floyd et al., 2000*). We targeted this denser ipsilateral PrL-P projection, and the targeting of left or right PrL-P pathways was counterbalanced among animals. The TNT experiments were similarly counterbalanced, and the tibial nerve in the hindlimb contralateral to the transfected PrL-P was ligated and transected, as has been done for similar investigations (*Lee et al., 2015*; *Huang et al., 2019*).

We wanted to investigate cortical control of the DPMS that routes via the PAG. The ventrolateral column of the PAG is a known source of descending pain modulation (*McMullan and Lumb, 2006a*; *McMullan and Lumb, 2006b*) and it is the caudal section that receives ascending inputs from the lumbar spinal cord that represents the hindpaws *Mouton et al., 1997*. In the rat, the caudal vlPAG is primarily innervated from rostral portions of the PrL (*Floyd et al., 2000*). Therefore, we targeted the delivery of CAV2-CMV-Cre to the caudal vlPAG and Cre-dependent AAV vectors to the rostral PrL on the same side.

A craniotomy was made over the PAG (Anteroposterior (AP) −7.5−8.5, Mediolateral (ML) ±1.8 mm). CAV2-CMV-CRE (300 nl/4.95×10$^8$ physical particles each site; Institut De Génétique

Moléculaire De Montpellier, France) was delivered to the vlPAG at two caudal sites; AP −7.5, ML 1.8, Dorsoventral (DV) 5.4 and AP −8.00, ML 1.8, DV 5.4 mm from the brain surface with a 9° lateral to medial angle. Approximately 20 nl of fluorescent microspheres (RetroBeads, Interchim, USA) were included in the injectant in some animals to mark the injection site.

A second craniotomy was made over mPFC (AP +3.0 to 5.0, ML ±0.4 to 0.8) allowing injection of Cre-dependent AAVs encoding ChR2 and hM4Di:

- AAV2-EF1a-DIO-hChR2-EYFP ($3.2 \times 10^{12}$ vg·ml$^{-1}$; UNC Vector Core, USA) and
- AAV2-hSyn-DIO-hM4Di-mCherry ($4.6 \times 10^{12}$ vg·ml$^{-1}$; Addgene, USA).

These were mixed to equal titres and delivered to the rostral PrL at three anteroposterior locations and at two dorsoventral levels.

- AP +4.2, ML ±0.6, DV −2.5 and −2.0 mm
- AP +3.8, ML ±0.6, DV −3.3 and −2.5 mm
- AP +3.20, ML ±0.6, DV −3.3 and −2.5 mm

Viral vectors were delivered using a pulled glass pipette (Broomall, USA) attached via silicon tubing to a 25-μl Hamilton syringe (Hamilton Company, USA). The whole system was filled with paraffin oil to allow for back filling of the pipette tip with viral vector. Delivery of the vector was controlled using a motorised syringe pump (Aladdin Syringe Pump, World Precision Instruments, USA), delivered at a rate of 200 nl/min and pipettes were left in place for ~10 min following vector delivery to allow for vector redistribution into the parenchyma.

An optic fibre (Lambda B, NA 0.66, length 4.4 mm, light-emitting length 2 mm, tapering from a width 200 to <5 μm at the tip, Optogenix, Italy) was inserted to AP +4.2, ML ±0.6, DV −3.3 mm from the cortical surface to enable light delivery across the full dorsoventral extent of the rostral PrL. Four skull screws were placed within separate cranial plates (M1, 1 mm diameter, 3 mm length; NewStar Fastenings, UK). The optic fibre was secured to an adjacent scull screw using Gentamicin CMW DePuy bone cement (DuPuy Synthes; Johnson and Johnson, USA). The craniotomies from the vector injections were then filled with artificial dura (duraGel; Cambridge Neurotech, UK), the skull's surface covered with bone cement and the skin incision closed using adsorbable suture (Vicryl; Ethicon Inc, Johnson and Johnson, USA) leaving the optic fibre ferrule connector protruding.

## Tibial nerve transection

Rats underwent recovery surgery for TNT to produce a neuropathic pain state (*Richardson, 2015*). This model was chosen for its gradual development of hypersensitivity as well as known contributions of DPMS (*Hughes et al., 2013*; *Hughes et al., 2015*). Briefly, rats had induction of anaesthesia using isoflurane (5% in $O_2$; Henry Schinn, UK) and maintained at a surgical plane of anaesthesia using 2–3% isoflurane in $O_2$. The tibial nerve contralateral to the transfected PrL-P pathway was exposed and transected before the wound closed. An incision was made from below the hip, parallel to the femur, and toward the knee. The underlying connective tissue was dissected away, and the fascial plane between gluteus superficialis and bicep femoris was dissected to expose the branches of the sciatic nerve. The Tibial nerve was identified and carefully freed from connective tissue. Two ligatures of sterile 5.0 braided silk (Fine Science Tools, Germany) were tightly ligated ~5 mm apart. The length of nerve between the two sutures was then transected and removed leaving the ligatures in place. The overlying muscle and skin were closed using adsorbable suture. Post-surgery, no analgesic was provided so as to not interfere with pain state development.

## Nociceptive testing

Rats underwent longitudinal nociceptive sensory testing before and after TNT. This was conducted with/without opto-activation and chemo-inhibition of PrL-P neurons to investigate their contribution to nociceptive threshold/pain-like behaviour in naive and TNT rats. Neuropathic animals underwent testing for mechanical (vF) before cold allodynia (acetone). There was more than 30 min between pre-CNO and post-CNO. All behaviours were recorded using a video camera (c930; Logitech, Switzerland) attached to a computer running video acquisition software (OBS Studio, Open Broadcaster Software) for offline analysis.

## Heat sensitivity

Thermal withdrawal latencies were measured for the hindpaw (*Hargreaves et al., 1988*). Animals were habituated to the testing apparatus and experimenter for 10 min for at least 5 days prior to the start of the experiment. On experimental days, animals were placed in a Plexiglass chamber on top of a raised glass plate so that the infrared (IR) beam (Ugo Basile Plantar test, Italy) could be positioned under the plantar surface of the hindpaws. The IR beam intensity was adjusted individually for each animal so that animals withdrew their paws at a latency of ~8 s (mean IR intensity=57±0.6). Animals had IR light delivered to both left and right hindpaw with ~4 min interstimulus interval between paws and hence >8 min interstimulus interval between consecutive stimuli on the same paw to prevent sensitisation. A cut-off latency of 15 s was used to prevent tissue damage and subsequent sensitisation. Stability of baseline withdrawal latency was considered to have been achieved when three consecutive latencies were within 2 s of each other.

## Punctate mechanical sensitivity

To assess mechanical sensitivity, animals were placed in a Plexiglass chamber on top of a raised metal grid to allow access to the plantar surface of the hindpaw. Rats were habituated to the testing apparatus and experimenter for 10 min at least 5 days before the start of the experiment. vF filaments (range 2.36–5.18 mN; Ugo Basile, Italy) were applied to the lateral aspect of the plantar surface of the hindpaw. The 50% withdrawal threshold was determined using the Massey-Dixon up-down method (*Chaplan et al., 1994*).

## Acetone

To assess cold sensitivity, rats were placed in a Plexiglass chamber on top of a raised metal grid to allow access to the plantar surface of the hindpaw. Using a 1-ml syringe, a drop of acetone (~0.1 ml) was applied to the lateral aspect of the hindpaw and the number of nocicfensive events (paw shakes, licks, and/or bites) recoded for up to 1 min following application using event logging software (*Friard and Gamba, 2016*). This was repeated three times for each paw with an ISI of 2 min.

## Manipulation of PrL-P neurons

For experiments involving opto-activation of PrL-P neurons Naive[PrL-P.ChR2:hM4Di], Naive[PrL-P.Control], TNT[PrL-P.ChR2:hM4Di], and TNT[PrL-P.Control] rats were tethered to a light source (445 nm diode laser; Omicron Laserage, Germany) using an optical fibre patch cable (FT200EMT; Thorlabs, USA) to connect the head-mounted ferrule to the laser source allowing blue light to be delivered to the PrL via the implanted optic fibre. Once stable baseline withdrawal latencies were obtained, two light stimulation rounds (445 nm, 10–15 mW, 20 hz, 10 ms pulse width, starting 1 min before initiation of the IR beam) and two no light rounds were delivered to the PrL in a randomised order. Output of optic fibres was determined prior to implant by measuring the light power at the fibre tip over a range of laser strengths using a monitor (PM120D; Thorlabs, USA). The average withdrawal latency for light stimulation rounds was compared to the average withdrawal latency for low-light stimulation rounds.

Experiments involving chemo-inhibition of PrL-P neurons were conducted on a separate day to opto-activation experiments. Once stable baseline withdrawal latencies for each hindpaw were obtained, animals received clozapine-N-oxide (CNO), the selective ligand for the hM4Di receptor, via an intraperitoneal injection at a dose of 2.5 mg·kg$^{-1}$. Animals were placed back in the testing box, and IR hindpaw stimulation started 10 min after CNO delivery. Withdrawal latencies to plantar IR stimulation were recorded for both hindpaws for at least 60 min post-injection, and the average withdrawal latency for recordings 20–40 min following CNO delivery were compared to the average baseline latencies for each paw.

The effect of chemo-inhibition of PrL-P neurons on mechanical withdrawal thresholds was assessed in TNT[PrL-P.ChR2-hM4Di] and TNT[PrL-P.Control] rats. Following, pre-CNO testing animals received systemic CNO (2.5 mg·kg$^{-1}$) via an i.p. injection and returned to their home cage. Twenty minutes following CNO injection, animals were placed back in the testing chamber and allowed to habituate for 10 min. vF testing was repeated at 30 min post-CNO. The 50% withdrawal threshold obtained following CNO was compared to pre-CNO withdrawal threshold for that day.

The effect of opto-activation of PrL-P neurons on the 50% withdrawal thresholds was assessed in TNT[PrL-P.ChR2-hM4Di] and TNT[PrL-P.Control] rats at a late state (>21 days). Rats underwent baseline vF

testing prior to blue light delivery as previously described. Then, blue light (445 nm, 10–15 mW, 20 hz, 10 ms pulse width) was delivered continuously starting 1 min prior to vF testing and continuing to the end of testing. The 50% withdrawal threshold of the ipsilateral (injured) paw was compared with and without opto-activation of PrL-P neurons.

## Behavioural testing

Conditioned place aversion: A second cohort of TNT rats that did not have longitudinal sensory testing was tested in a conditioned place aversion paradigm to assess the contributions of PrL-P neurons to affective state. TNT$^{PrL-P.ChR2:hM4Di}$ and TNT$^{PrL-P.Control}$ rats were habituated to a three-compartment box with a neutral central compartment connecting two larger conditioning chambers. Chambers differed in their visual and tactile cues ('bars' or 'holes' for flooring and vertically or horizontally striped wallpaper with equal luminosity) to maximise their differentiation. A Baslar camera (acA1300-60 gm) with a varifocal lens (Computar H3Z4512CS-IR) connected to EthovisionXT (Noldus, NL) was used to record the time rats spent in each compartment. Rats were allowed to freely explore all three compartments for 20 min on day 1 to obtain baseline preference. No rats exhibited excessive chamber bias (>80% total time in a single chamber). After habituation, rats had TNT surgery and 2 days later started conditioning sessions (over 4 days) in which a compartment was paired with CNO (2.5 mg·kg$^{-1}$ i.p.) or vehicle (two sessions each). The chamber–drug pairings and the order in which they were presented were randomised and counterbalanced among animals. For each pairing session, rats received CNO or vehicle and were returned to their home cage for 10 min to prevent any negative association between restraint/injection and conditioning compartment. Rats were then placed in the conditioning compartment for 35 min. A single pairing session was conducted on each of the 4 days to prevent carry over of any CNO effects. Pairing sessions for each rat were conducted at the same time on each day. On the test day, animals were placed in the neutral compartment and allowed to freely explore all three compartments for a total of 20 min and the time spent in each compartment recorded. A 'preference score' was calculated by taking the percentage of time spent in the CNO-paired compartment on the test day (relative to the total time spent in all three chambers), normalised by the percentage time spent in the same chamber on pre-test day (relative to total time spent in all three chambers; *Meda et al., 2019*). Preference scores for CNO- and vehicle-paired chambers were compared within each animal. Preference or aversion to CNO-paired chamber is expected to be influenced by the valence of chemo-inhibition of PrL-P neurons. Preference scores of <1 indicate place aversion and those >1 indicate preference.

## Electrophysiology

In vivo spinal dorsal horn recordings: TNT$^{PrL-P.ChR2:hM4Di}$ rats were terminally anaesthetised with urethane (1.2–2 g·kg$^{-1}$ i.p., Sigma). The spinal cord was exposed by laminectomy over T13–L3 spinal segments to allow access to the spinal cord (*Leith et al., 2014*; *Drake et al., 2016*). The animal was placed in a stereotaxic frame with spinal clamps (Narishige, Japan) and the spinal cord stabilised at T12 and L4, and a bath formed by skin elevation. A reference electrode was placed in nearby musculature. The spinal dura matter was carefully removed using bent-tipped needles (25G) under binocular vision. The skin pool was filled with warm agar and, once cool, a recording window cut out and the void filled with warm (~37°C) mineral oil. Using a hydraulic manipulator (Narishige, Japan), a four contact silicon probe (Q-probe; NeuroNexus, USA) was advanced into the spinal dorsal horn and recordings of single dorsal horn neurons made between 250 and 800 µm deep to the surface. Neural activity was amplified and digitised on a headstage microchip (RHD2132; Intan technology) and captured to computer at 30 kHz using an Open Ephys acquisition system and associated software (OpenEphys, USA).

Low threshold brush and touch applied to the paw were used as a search stimulus as the recording electrode was advanced into the spinal dorsal horn. Once single units were isolated, non-noxious and/or noxious mechanical (vF filaments) and cold (acetone) were applied to the receptive field on the lateral aspect of the ipsilateral hind leg/paw. WDR neurons were identified by their graded response to non-noxious and noxious stimuli (≥15 g vF). A baseline stimulus–response relationship was obtained by applying 4 and 15 g vF filaments and a single drop of acetone to the receptive field. This was repeated three times for each stimulus with a 10 s inter-stimulus interval between vF filaments and 1 min between acetone drops. To optogenetically activate the PrL-P neurons, blue

light was delivered via the implanted optic fibre (445 nm, 10–15 mW, 20 Hz, 10 ms pulse width) continuously starting 1 min prior to peripheral stimulation and lasting until the end of the stimulus set (4 and 15 g vF hairs and acetone stimuli were reapplied three times). The average number of evoked action potentials for each stimulus was compared before, during, and 5 mins following opto-activation of PrL-P neurons.

## Histological processing

Tissue collection and processing: Rats were killed with an overdose of pentobarbital (20 mg/100 g, i. p., Euthalal, Merial Animal Health) and perfused trans-cardially with 0.9% NaCl (1 ml/g) followed by 4% formaldehyde in phosphate buffer (PB). The brains were dissected and post-fixed overnight in the same solution before cryoprotection in 30% sucrose in PB. Coronal sections were cut at 40 µm using a freezing microtome and left free floating for fluorescent immunohistochemistry or mounted on slides to identify optic fibre tracts and/or injection sites for viral vector delivery using light microscopy.

## Immunofluorescence

Tissue sections were incubated free floating on a shaking platform with PB containing 0.3% Triton-X100, 5% normal goat serum (Sigma), and primary antibodies to detect EGFP (ab13970, Abcam) or mCherry (5993–100, BioVision) for 24 hr at room temperature. After washing with PB, sections were incubated for 3 hr at room temperature with an appropriate Alexa Fluor secondary antibody. Then, sections were washed before mounting on glass slides in 1% gelatin solution and, once dried, cover slipped using FluroSave reagent (345789; Merck-Millipore, Germany). Sections were imaged on a Leica DM16000 inverted epifluorescence microscope equipped with Leica DFC365FX digital camera and LAS-X acquisition software.

## Transduction mapping

To create maps of the distribution of transfected neurons within the mPFC, a series of coronal mPFC sections from three animals were manually plotted. Each section was paired to a matching coronal diagram from the Rat Brain Atlas (*Paxinos and Watson, 2007*), at ~120 µm intervals (every third section). Using an epiflurescent microscope (Zeiss Axioskop II inverted microscope equipped with a CooLED pE-100 excitation system, filter blocks – red: filter set number 15 [DM 580 nm, BP 546/12 nm, LP 590 nm] and green: filter set number 09 [DM 510 nm, BP 450–490 nm, LP 515 nm]), mCherry + cells were plotted. The diagrams were digitised into the photo editing software GIMP.2 (Creative Commons), allowing superimposition to create conjunction maps indicating the extent of labelled areas of the mPFC within each cohort. A digital grid was used to divide up the cortical field and the number of positively labelled neurons counted within each 0.2 mm$^2$ grid from each animal. The consistency of positively labelled neurons within each grid square was represented on a grayscale with black indicating positively labelled cells in all rats and white indicating no cells. To determine the proportion of transfected neurons that co-expressed both ChR2-EYFP and hM4Di-mCherry composite widefield images were taken at 20× magnification of every sixth section in a series of consecutive mPFC section from ~+5.10 to +2.8 mm from bregma and from three experimental animals. From these images, the distribution of EYFP, mCherry, and colocalised neurons were quantified.

## Drugs

ClozepineN-Oxide (Tocris, UK) was purchased and made up on the day of use in Dimethyl sulfoxide (DMSO) and diluted with 0.9% NaCl to a final concentration of 2.5 mg·ml$^{-1}$ and 5% DMSO.

## Quantification and statistics

All statistical analyses were conducted using GraphPad Prism 8. All data are presented as mean ± standard error of mean (SEM). Sample sizes were calculated using online power calculators with alpha set at 0.05, power >0.9 and using effect sizes and sample variation estimated from previous experience and with reference to literature (*Hughes et al., 2013*; *Lee et al., 2015*; *Zhang et al., 2015*; *Drake et al., 2016*; *Hirschberg et al., 2017*). Student's t-test (paired and unpaired), RM one- and two-way ANOVAs, or mixed model were used to compare groups as appropriate. This mixed model uses a compound symmetry covariance matrix and is fit using restricted maximum likelihood

(REML). Sidak's or Dunnett's post-test was used for comparisons between multiple groups and where appropriate. The number of replications (n) is the number of data points used in the statistical test that is either the number of animals for behavioural testing or the number of neurons for electrophysiological experiments.

## Acknowledgements

We would like to thank Mrs Rachel Bissett for her assistance with histological processing and the Wolfson Bioimaging Facility for their support and assistance with image acquisition. We would like to thank Dr Eric J Kremer for his kind gift of CAV-CMV-CRE, and Dr Brian Roth and Dr Karl Deisseroth for the supply of viral vectors used in this work. This work was supported by the Medical Research Council Grant number MR/P00668/X1.

## Additional information

### Funding

| Funder | Grant reference number | Author |
| --- | --- | --- |
| Medical Research Council | MR/P00668/X1 | Robert A R Drake<br>Richard Apps<br>Bridget M Lumb<br>Anthony E Pickering |

The funders had no role in study design, data collection and interpretation, or the decision to submit the work for publication.

### Author contributions

Robert AR Drake, Conceptualization, Formal analysis, Supervision, Funding acquisition, Validation, Investigation, Visualization, Methodology, Writing - original draft, Project administration, Writing - review and editing; Kenneth A Steel, Investigation, Visualization; Richard Apps, Resources, Supervision, Funding acquisition, Writing - review and editing; Bridget M Lumb, Conceptualization, Resources, Supervision, Funding acquisition, Writing - review and editing; Anthony E Pickering, Conceptualization, Resources, Formal analysis, Supervision, Funding acquisition, Visualization, Writing - review and editing

### Author ORCIDs

Robert AR Drake (iD) https://orcid.org/0000-0003-2381-7198
Bridget M Lumb (iD) http://orcid.org/0000-0002-0268-6419
Anthony E Pickering (iD) http://orcid.org/0000-0003-0345-0456

### Ethics

Animal experimentation: All experimental and surgical procedures were conducted in accordance with the UK Animals (Scientific Procedures) Act (1998) and local Animal Welfare and Ethical Review Body (AWERB).

### Decision letter and Author response

Decision letter https://doi.org/10.7554/eLife.65156.sa1
Author response https://doi.org/10.7554/eLife.65156.sa2

## Additional files

### Supplementary files

- Transparent reporting form

### Data availability

All data generated or analysed are included in the manuscript.

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
