## [Decision Letter]

**Acceptance summary:**

Using optogenetic and chemogenetic approaches the authors demonstrate that loss of cortical drive from prelimbic cortex to a periaqueductal gray-engaged descending pain modulatory system underpins, at least in part, the expression of neuropathic sensitization after nerve injury.

**Decision letter after peer review:**

Thank you for submitting your article "Loss of cortical control over the descending pain modulatory system determines pain state development in rats" for consideration by *eLife*. Your article has been reviewed by three peer reviewers, and the evaluation has been overseen by a Reviewing Editor and Kate Wassum as the Senior Editor. The following individual involved in review of your submission has agreed to reveal their identity: Mary Heinricher (Reviewer #3).

The reviewers have discussed the reviews with one another and the Reviewing Editor has drafted this decision to help you prepare a revised submission.

Using optogenetic and chemogenetic approaches, the present paper builds on and extends earlier work by demonstrating that the loss of cortical drive (from prelimbic cortex) to the periacqueductal grey-engaged descending pain modulatory system underpins, at least in part, the expression of neuropathic sensitization after nerve injury. The paper, on the whole, is cogently written and easy to follow, and the analysis of the efficacy of techniques used is thorough.

However, there are minor concerns that should be addressed in a revised manuscript:

1) In the Materials and methods section; the authors state "The IR beam intensity was adjusted so that animals withdrew their paws at a latency of ~8s." Please include whether this intensity was adjusted separately per each animal's response, or if it was averaged across animals, as well as what the absolute value of IR intensity was.

2) For Figure 3F, "For the contralateral (uninjured) paw PrL-P chemo-inhibition significantly reduced mechanical withdrawal thresholds at day 3 post-TNT from 13.5±0.7 to 9.0±1.4 grams (2Way ANOVA. CNO F(1,30) = 5.77 p=0.02, Sidak's post-test day 3 pre-CNO vs. CNO p=0.03, n=16) but not thereafter (Figure 3F)." – since CNO alone did not reduce withdrawal thresholds in naive animals, please include that this may indicate the effect of injury (and possibly decussating projections) on sensitivity in the contralateral paw.

3) In the Introduction section, there is a typographical error in the following sentence: "Following expression of actuator (n=24) and affective aspects (n=20) of pain like behaviour in neuropathic (TNT^PrL-P.ChR2-hM4Di^ and TNT^PrL-P.Control^) and uninjured (Naive^PrL-P.ChR2-hM4Di^ and Naive^PrL-P.Control^) rats."

4) The authors provide the numbers of rats that were excluded from each group due to various experimental issues; this is much appreciated. However, a supplementary table or short insert into the text providing the exact number of animals used per group per experimental paradigm would also be very useful for readers.

5) Discussion, sixth paragraph: revise to “overlooked”.

6) Can the authors be certain that the presence of an optic fiber in a region adjacent to the PrL (presumably either the ACC or IL), and the resulting mechanical damage or alteration in neuronal activity associated with this, itself had no effect on rat behaviour in the chemo-inhibition experiments? Some discussion of potential off target effects should be included.

7) The authors state "PrL-P projections extend bilaterally from each hemisphere with the ipsilateral projection bring denser that the contralateral projection (~60 v 40% of total labelled cells from retrograde tracing (Floyd et al., 2000))." They need to make it clear here which side (i.e. left or right) they are referring to with the term “ipsilateral” (presumably relative to the side of nerve injury but this should be clarified explicitly here and left or right equated to ipsilateral/contralateral.

8) The authors state in a few places that "Activation of PrL-P neurons produces analgesia in neuropathic rats by inhibition at a spinal level". The use of the term “analgesia” is usually best reserved for clinical/human studies and might be better replaced with antinociceptive, antiallodynic or antihyperalgesic.

9) The Discussion is well-written overall. The authors state "it is the loss of cortical drive to the descending pain modulatory system that underpins the expression of neuropathic sensitisation after nerve injury." I think their results demonstrate that this is likely to be a mechanism that underpins such expression, but their data don't confirm that it is the only mechanism. Thus, the authors should revise/tone down this phraseology.

---

## [Author Response]

[…] There are minor concerns that should be addressed in a revised manuscript:1) In the Materials and methods section; the authors state "The IR beam intensity was adjusted so that animals withdrew their paws at a latency of ~8s." Please include whether this intensity was adjusted separately per each animal's response, or if it was averaged across animals, as well as what the absolute value of IR intensity was.

The intensity was adjusted for each animal individually and this has now been stated clearly and a mean intensity value ± SEM provided (subsection “Stereotaxic injection/implants”).

2) For Figure 3F, "For the contralateral (uninjured) paw PrL-P chemo-inhibition significantly reduced mechanical withdrawal thresholds at day 3 post-TNT from 13.5±0.7 to 9.0±1.4 grams (2Way ANOVA. CNO F(1,30) = 5.77 p=0.02, Sidak's post-test day 3 pre-CNO vs. CNO p=0.03, n=16) but not thereafter (Figure 3F)." – since CNO alone did not reduce withdrawal thresholds in naive animals, please include that this may indicate the effect of injury (and possibly decussating projections) on sensitivity in the contralateral paw.

We agree and have added the following text to the Discussion to address this point:

“Notably, in addition to affecting the sensitivity of the injured paw, antagonism of spinal alpha-2-receptors reveals contralateral allodynia (Hughes et al., 2013), something we too observe with chemoinhibition of PrL-P neurones. This indicates that nerve injury drives central sensitisation that affects the response to nociceptive inputs from the uninjured paw to produce contralateral hypersensitivity that can be revealed by blockade of the descending modulation of spinal processing originating from the pre-limbic cortex.”

3) In the Introduction section, there is a typographical error in the following sentence: "Following expression of actuator (n=24) and affective aspects (n=20) of pain like behaviour in neuropathic (TNT^PrL-P.ChR2-hM4Di^ and TNT^PrL-P.Control^) and uninjured (Naive^PrL-P.ChR2-hM4Di^ and Naive^PrL-P.Control^) rats."

This has now been corrected:

“Following expression of actuator proteins, the effect of selective optoactivation and chemoinhibition of PrL-P neurons on sensory (n=24) and affective aspects (n=20) of pain like behaviour was assessed in neuropathic (TNT^PrL-P.ChR2-hM4Di^ and TNT^PrL-P.Control^) and uninjured (Naive^PrL-P.ChR2-hM4Di^ and Naive^PrL-P.Control^) rats.”

4) The authors provide the numbers of rats that were excluded from each group due to various experimental issues; this is much appreciated. However, a supplementary table or short insert into the text providing the exact number of animals used per group per experimental paradigm would also be very useful for readers.

This information has now been included in a table in the Materials and methods section.

5) Discussion, sixth paragraph: revise to “overlooked”.

This has now been corrected (Discussion).

6) Can the authors be certain that the presence of an optic fibere in a region adjacent to the PrL (presumably either the ACC or IL), and the resulting mechanical damage or alteration in neuronal activity associated with this, itself had no effect on rat behaviour in the chemo-inhibition experiments? Some discussion of potential off target effects should be included.

The following addition has been made to the Discussion:

“A tapered optic fibere (tapering over a 2mm length from 200µm to a diameter of <5µm) was inserted into the PrL to allow light delivery for optoactivation experiments. […] Our data demonstrate that rats implanted with tapered optic fibres are still able to exhibit aversive place conditioning indicating that any cortical disruption produced by our fibres does not impact CG1/ACC function relevant our investigation.

7) The authors state "PrL-P projections extend bilaterally from each hemisphere with the ipsilateral projection bring denser that the contralateral projection (~60 v 40% of total labelled cells from retrograde tracing (Floyd et al., 2000))." They need to make it clear here which side (i.e. left or right) they are referring to with the term “ipsilateral” (presumably relative to the side of nerve injury but this should be clarified explicitly here and left or right equated to ipsilateral/contralateral.

The statement above is accurate and we have attempted to make the text clearer. We transfected the ipsilateral projection between the prelimbic cortex and the periaqueductal grey, and we counterbalanced the experiments by transfecting either the left or right PrL-P and applying stimuli to the ipsilateral or contralateral hind paw. Therefore, experiments conducted in uninjured animals, in which the effects of pathway activation/inhibition on hind paw withdrawal thresholds were determined, ipsilateral and contralateral refers to the hind paw with respect to the side of transfected PrL-P projection. This is clearly stated in the figure panel above the relevant data (Figure 2B and C).

In TNT animals we again counterbalanced (left and right) and ligated the tibial nerve contralateral to the transfected pathway (as has been done in similar investigations), and this is clearly stated in the relevant Materials and methods section. In experiments conducted in TNT animals ipsilateral and contralateral refer to injured and uninjured paw (as is common in pain literature) and this is clearly stated in the relevant figure panels (Figure 3E-J).

To further clarify for the reader, we have added in this information earlier in the Materials and methods section to bring all the relevant information together in one place.

8) The authors state in a few places that "Activation of PrL-P neurons produces analgesia in neuropathic rats by inhibition at a spinal level". The use of the term “analgesia” is usually best reserved for clinical/human studies and might be better replaced with antinociceptive, antiallodynic or antihyperalgesic.

We have amended the text to the following:

“Restoration of PrL-P neuronal tone attenuated allodynia in established neuropathic sensitisation.” and “PrL-P produces antinociception in neuropathic pain by inhibition of dorsal horn nociceptive responses.”

9) The Discussion is well-written overall. The authors state "it is the loss of cortical drive to the descending pain modulatory system that underpins the expression of neuropathic sensitisation after nerve injury." I think their results demonstrate that this is likely to be a mechanism that underpins such expression, but their data don't confirm that it is the only mechanism. Thus, the authors should revise/tone down this phraseology.

We have now revised this statement to the following:

“We postulate that this cortical – midbrain – spinal network allows executive control of nociception and regulation of pain, and that the loss of cortical drive to the descending pain modulatory system is a major contributor to the expression of neuropathic sensitisation after nerve injury.”